# Global Analysis of Expectation Maximization for Mixtures of Two Gaussians

**Ji Xu**
Columbia University
jixu@cs.columbia.edu

**Daniel Hsu**
Columbia University
djhsu@cs.columbia.edu

**Arian Maleki**
Columbia University
arian@stat.columbia.edu

## Abstract

Expectation Maximization (EM) is among the most popular algorithms for estimating parameters of statistical models. However, EM, which is an iterative algorithm based on the maximum likelihood principle, is generally only guaranteed to find stationary points of the likelihood objective, and these points may be far from any maximizer. This article addresses this disconnect between the statistical principles behind EM and its algorithmic properties. Specifically, it provides a global analysis of EM for specific models in which the observations comprise an i.i.d. sample from a mixture of two Gaussians. This is achieved by (i) studying the sequence of parameters from idealized execution of EM in the infinite sample limit, and fully characterizing the limit points of the sequence in terms of the initial parameters; and then (ii) based on this convergence analysis, establishing statistical consistency (or lack thereof) for the actual sequence of parameters produced by EM.

## 1 Introduction

Since Fisher's 1922 paper (Fisher, 1922), maximum likelihood estimators (MLE) have become one of the most popular tools in many areas of science and engineering. The asymptotic consistency and optimality of MLEs have provided users with the confidence that, at least in some sense, there is no better way to estimate parameters for many standard statistical models. Despite its appealing properties, computing the MLE is often intractable. Indeed, this is the case for many *latent variable models* $\{f(\mathcal{Y}, \boldsymbol{z}; \boldsymbol{\eta})\}$, where the *latent variables* $\boldsymbol{z}$ are not observed. For each setting of the parameters $\boldsymbol{\eta}$, the marginal distribution of the observed data $\mathcal{Y}$ is (for discrete $\boldsymbol{z}$)

$$f(\mathcal{Y}; \boldsymbol{\eta}) = \sum_{\boldsymbol{z}} f(\mathcal{Y}, \boldsymbol{z}; \boldsymbol{\eta}).$$

It is this marginalization over latent variables that typically causes the computational difficulty. Furthermore, many algorithms based on the MLE principle are only known to find stationary points of the likelihood objective (e.g., local maxima), and these points are not necessarily the MLE.

### 1.1 Expectation Maximization

Among the algorithms mentioned above, Expectation Maximization (EM) has attracted more attention for the simplicity of its iterations, and its good performance in practice (Dempster et al., 1977; Redner and Walker, 1984). EM is an iterative algorithm for climbing the likelihood objective starting from an initial setting of the parameters $\hat{\boldsymbol{\eta}}^{\langle 0 \rangle}$. In iteration $t$, EM performs the following steps:

$$\text{E-step:} \qquad \hat{Q}(\boldsymbol{\eta} \mid \hat{\boldsymbol{\eta}}^{\langle t \rangle}) \triangleq \sum_{\boldsymbol{z}} f(\boldsymbol{z} \mid \mathcal{Y}; \hat{\boldsymbol{\eta}}^{\langle t \rangle}) \log f(\mathcal{Y}, \boldsymbol{z}; \boldsymbol{\eta}), \qquad (1)$$

$$\text{M-step:} \qquad \hat{\boldsymbol{\eta}}^{\langle t+1 \rangle} \triangleq \arg\max_{\boldsymbol{\eta}} \hat{Q}(\boldsymbol{\eta} \mid \hat{\boldsymbol{\eta}}^{\langle t \rangle}), \qquad (2)$$

In many applications, each step is intuitive and can be performed very efficiently.

Despite the popularity of EM, as well as the numerous theoretical studies of its behavior, many important questions about its performance—such as its convergence rate and accuracy—have remained unanswered. The goal of this paper is to address these questions for specific models (described in Section 1.2) in which the observation $\mathcal{Y}$ is an i.i.d. sample from a mixture of two Gaussians.

Towards this goal, we study an idealized execution of EM in the large sample limit, where the E-step is modified to be computed over an infinitely large i.i.d. sample from a Gaussian mixture distribution in the model. In effect, in the formula for $\hat{Q}(\boldsymbol{\eta} \mid \hat{\boldsymbol{\eta}}^{\langle t \rangle})$, we replace the observed data $\mathcal{Y}$ with a random variable $\boldsymbol{Y} \sim f(\boldsymbol{y}; \boldsymbol{\eta}^\star)$ for some Gaussian mixture parameters $\boldsymbol{\eta}^\star$ and then take its expectation. The resulting E- and M-steps in iteration $t$ are

$$\text{E-step:} \qquad Q(\boldsymbol{\eta} \mid \boldsymbol{\eta}^{\langle t \rangle}) \triangleq \mathbb{E}_{\boldsymbol{Y}} \left[ \sum_{\boldsymbol{z}} f(\boldsymbol{z} \mid \boldsymbol{Y}; \boldsymbol{\eta}^{\langle t \rangle}) \log f(\boldsymbol{Y}, \boldsymbol{z}; \boldsymbol{\eta}) \right], \qquad (3)$$

$$\text{M-step:} \qquad \boldsymbol{\eta}^{\langle t+1 \rangle} \triangleq \arg\max_{\boldsymbol{\eta}} Q(\boldsymbol{\eta} \mid \boldsymbol{\eta}^{\langle t \rangle}). \qquad (4)$$

This sequence of parameters $(\boldsymbol{\eta}^{\langle t \rangle})_{t \geq 0}$ is fully determined by the initial setting $\boldsymbol{\eta}^{\langle 0 \rangle}$. We refer to this idealization as *Population EM*, a procedure considered in previous works of Srebro (2007) and Balakrishnan et al. (2014). Not only does Population EM shed light on the dynamics of EM in the large sample limit, but it can also reveal some of the fundamental limitations of EM. Indeed, if Population EM cannot provide an accurate estimate for the parameters $\boldsymbol{\eta}^\star$, then intuitively, one would not expect the EM algorithm with a finite sample size to do so either. (To avoid confusion, we refer the original EM algorithm run with a finite sample as *Sample-based EM*.)

## 1.2 Models and Main Contributions

In this paper, we study EM in the context of two simple yet popular and well-studied Gaussian mixture models. The two models, along with the corresponding Sample-based EM and Population EM updates, are as follows:

**Model 1.** The observation $\mathcal{Y}$ is an i.i.d. sample from the mixture distribution $0.5N(-\boldsymbol{\theta}^\star, \boldsymbol{\Sigma}) + 0.5N(\boldsymbol{\theta}^\star, \boldsymbol{\Sigma})$; $\boldsymbol{\Sigma}$ is a known covariance matrix in $\mathbb{R}^d$, and $\boldsymbol{\theta}^\star$ is the unknown parameter of interest.

1. Sample-based EM iteratively updates its estimate of $\boldsymbol{\theta}^\star$ according to the following equation:

$$\hat{\boldsymbol{\theta}}^{\langle t+1 \rangle} = \frac{1}{n} \sum_{i=1}^{n} \left( 2\mathsf{w}_d \left( \boldsymbol{y}_i, \hat{\boldsymbol{\theta}}^{\langle t \rangle} \right) - 1 \right) \boldsymbol{y}_i, \qquad (5)$$

where $\boldsymbol{y}_1, \ldots, \boldsymbol{y}_n$ are the independent draws that comprise $\mathcal{Y}$,

$$\mathsf{w}_d(\boldsymbol{y}, \boldsymbol{\theta}) \triangleq \frac{\phi_d(\boldsymbol{y} - \boldsymbol{\theta})}{\phi_d(\boldsymbol{y} - \boldsymbol{\theta}) + \phi_d(\boldsymbol{y} + \boldsymbol{\theta})},$$

and $\phi_d$ is the density of a Gaussian random vector with mean $\boldsymbol{0}$ and covariance $\boldsymbol{\Sigma}$.

2. Population EM iteratively updates its estimate according to the following equation:

$$\boldsymbol{\theta}^{\langle t+1 \rangle} = \mathbb{E}(2\mathsf{w}_d(\boldsymbol{Y}, \boldsymbol{\theta}^{\langle t \rangle}) - 1)\boldsymbol{Y}, \qquad (6)$$

where $\boldsymbol{Y} \sim 0.5N(-\boldsymbol{\theta}^\star, \boldsymbol{\Sigma}) + 0.5N(\boldsymbol{\theta}^\star, \boldsymbol{\Sigma})$.

**Model 2.** The observation $\mathcal{Y}$ is an i.i.d. sample from the mixture distribution $0.5N(\boldsymbol{\mu}_1^\star, \boldsymbol{\Sigma}) + 0.5N(\boldsymbol{\mu}_2^\star, \boldsymbol{\Sigma})$. Again, $\boldsymbol{\Sigma}$ is known, and $(\boldsymbol{\mu}_1^\star, \boldsymbol{\mu}_2^\star)$ are the unknown parameters of interest.

1. Sample-based EM iteratively updates its estimate of $\boldsymbol{\mu}_1^\star$ and $\boldsymbol{\mu}_2^\star$ at every iteration according to the following equations:

$$\hat{\boldsymbol{\mu}}_1^{\langle t+1 \rangle} = \frac{\sum_{i=1}^{n} \mathsf{v}_d(\boldsymbol{y}_i, \hat{\boldsymbol{\mu}}_1^{\langle t \rangle}, \hat{\boldsymbol{\mu}}_2^{\langle t \rangle}) \boldsymbol{y}_i}{\sum_{i=1}^{n} \mathsf{v}_d(\boldsymbol{y}_i, \hat{\boldsymbol{\mu}}_1^{\langle t \rangle}, \hat{\boldsymbol{\mu}}_2^{\langle t \rangle})}, \qquad (7)$$

$$\hat{\boldsymbol{\mu}}_2^{\langle t+1 \rangle} = \frac{\sum_{i=1}^{n} (1 - \mathsf{v}_d(\boldsymbol{y}_i, \hat{\boldsymbol{\mu}}_1^{\langle t \rangle}, \hat{\boldsymbol{\mu}}_2^{\langle t \rangle})) \boldsymbol{y}_i}{\sum_{i=1}^{n} (1 - \mathsf{v}_d(\boldsymbol{y}_i, \hat{\boldsymbol{\mu}}_1^{\langle t \rangle}, \hat{\boldsymbol{\mu}}_2^{\langle t \rangle}))}, \qquad (8)$$

where $\boldsymbol{y}_1, \ldots, \boldsymbol{y}_n$ are the independent draws that comprise $\mathcal{Y}$, and

$$\mathsf{v}_d(\boldsymbol{y}, \boldsymbol{\mu}_1, \boldsymbol{\mu}_2) \triangleq \frac{\phi_d(\boldsymbol{y} - \boldsymbol{\mu}_1)}{\phi_d(\boldsymbol{y} - \boldsymbol{\mu}_1) + \phi_d(\boldsymbol{y} - \boldsymbol{\mu}_2)}.$$

2. Population EM iteratively updates its estimates according to the following equations:

$$\boldsymbol{\mu}_1^{\langle t+1 \rangle} = \frac{\mathbb{E}\mathsf{v}_d(\boldsymbol{Y}, \boldsymbol{\mu}_1^{\langle t \rangle}, \boldsymbol{\mu}_2^{\langle t \rangle})\boldsymbol{Y}}{\mathbb{E}\mathsf{v}_d(\boldsymbol{Y}, \boldsymbol{\mu}_1^{\langle t \rangle}, \boldsymbol{\mu}_2^{\langle t \rangle})}, \tag{9}$$

$$\boldsymbol{\mu}_2^{\langle t+1 \rangle} = \frac{\mathbb{E}(1 - \mathsf{v}_d(\boldsymbol{Y}, \boldsymbol{\mu}_1^{\langle t \rangle}, \boldsymbol{\mu}_2^{\langle t \rangle}))\boldsymbol{Y}}{\mathbb{E}(1 - \mathsf{v}_d(\boldsymbol{Y}, \boldsymbol{\mu}_1^{\langle t \rangle}, \boldsymbol{\mu}_2^{\langle t \rangle}))}, \tag{10}$$

where $\boldsymbol{Y} \sim 0.5N(\boldsymbol{\mu}_1^\star, \boldsymbol{\Sigma}) + 0.5N(\boldsymbol{\mu}_2^\star, \boldsymbol{\Sigma})$.

Our main contribution in this paper is a new characterization of the stationary points and dynamics of EM in both of the above models.

1. We prove convergence for the sequence of iterates for Population EM from each model: the sequence $(\boldsymbol{\theta}^{\langle t \rangle})_{t \geq 0}$ converges to either $\boldsymbol{\theta}^\star$, $-\boldsymbol{\theta}^\star$, or $\boldsymbol{0}$; the sequence $((\boldsymbol{\mu}_1^{\langle t \rangle}, \boldsymbol{\mu}_2^{\langle t \rangle}))_{t \geq 0}$ converges to either $(\boldsymbol{\mu}_1^\star, \boldsymbol{\mu}_2^\star)$, $(\boldsymbol{\mu}_2^\star, \boldsymbol{\mu}_1^\star)$, or $((\boldsymbol{\mu}_1^\star + \boldsymbol{\mu}_2^\star)/2, (\boldsymbol{\mu}_1^\star + \boldsymbol{\mu}_2^\star)/2)$. We also fully characterize the initial parameter settings that lead to each limit point.

2. Using this convergence result for Population EM, we also prove that the limits of the Sample-based EM iterates converge in probability to the unknown parameters of interest, as long as Sample-based EM is initialized at points where Population EM would converge to these parameters as well.

Formal statements of our results are given in Section 2.

## 1.3 Background and Related Work

The EM algorithm was formally introduced by Dempster et al. (1977) as a general iterative method for computing parameter estimates from incomplete data. Although EM is billed as a procedure for maximum likelihood estimation, it is known that with certain initializations, the final parameters returned by EM may be far from the MLE, both in parameter distance and in log-likelihood value (Wu, 1983). Several works characterize convergence of EM to stationary points of the log-likelihood objective under certain regularity conditions (Wu, 1983; Tseng, 2004; Vaida, 2005; Chrétien and Hero, 2008). However, these analyses do not distinguish between global maximizers and other stationary points (except, e.g., when the likelihood function is unimodal). Thus, as an optimization algorithm for maximizing the log-likelihood objective, the "worst-case" performance of EM is somewhat discouraging.

For a more optimistic perspective on EM, one may consider a "best-case" analysis, where (i) the data are an iid sample from a distribution in the given model, (ii) the sample size is sufficiently large, and (iii) the starting point for EM is sufficiently close to the parameters of the data generating distribution. Conditions (i) and (ii) are ubiquitous in (asymptotic) statistical analyses, and (iii) is a generous assumption that may be satisfied in certain cases. Redner and Walker (1984) show that in such a favorable scenario, EM converges to the MLE almost surely for a broad class of mixture models. Moreover, recent work of Balakrishnan et al. (2014) gives non-asymptotic convergence guarantees in certain models; importantly, these results permit one to quantify the accuracy of a pilot estimator required to effectively initialize EM. Thus, EM may be used in a tractable two-stage estimation procedures given a first-stage pilot estimator that can be efficiently computed.

Indeed, for the special case of Gaussian mixtures, researchers in theoretical computer science and machine learning have developed efficient algorithms that deliver the highly accurate parameter estimates under appropriate conditions. Several of these algorithms, starting with that of Dasgupta (1999), assume that the means of the mixture components are *well-separated*—roughly at distance either $d^\alpha$ or $k^\beta$ for some $\alpha, \beta > 0$ for a mixture of $k$ Gaussians in $\mathbb{R}^d$ (Dasgupta, 1999; Arora and Kannan, 2005; Dasgupta and Schulman, 2007; Vempala and Wang, 2004; Kannan et al., 2008; Achlioptas and McSherry, 2005; Chaudhuri and Rao, 2008; Brubaker and Vempala, 2008; Chaudhuri et al., 2009a). More recent work employs the method-of-moments, which permit the means of the

mixture components to be arbitrarily close, provided that the sample size is sufficiently large (Kalai et al., 2010; Belkin and Sinha, 2010; Moitra and Valiant, 2010; Hsu and Kakade, 2013; Hardt and Price, 2015). In particular, Hardt and Price (2015) characterize the information-theoretic limits of parameter estimation for mixtures of two Gaussians, and that they are achieved by a variant of the original method-of-moments of Pearson (1894).

Most relevant to this paper are works that specifically analyze EM (or variants thereof) for Gaussian mixture models, especially when the mixture components are well-separated. Xu and Jordan (1996) show favorable convergence properties (akin to super-linear convergence near the MLE) for well-separated mixtures. In a related but different vein, Dasgupta and Schulman (2007) analyze a variant of EM with a particular initialization scheme, and proves fast convergence to the true parameters, again for well-separated mixtures in high-dimensions. For mixtures of two Gaussians, it is possible to exploit symmetries to get sharper analyses. Indeed, Chaudhuri et al. (2009b) uses these symmetries to prove that a variant of Lloyd's algorithm (MacQueen, 1967; Lloyd, 1982) (which may be regarded as a hard-assignment version of EM) very quickly converges to the subspace spanned by the two mixture component means, without any separation assumption. Lastly, for the specific case of our Model 1, Balakrishnan et al. (2014) proves linear convergence of EM (as well as a gradient-based variant of EM) when started in a sufficiently small neighborhood around the true parameters, assuming a minimum separation between the mixture components. Here, the permitted size of the neighborhood grows with the separation between the components, and a recent result of Klusowski and Brinda (2016) quantitatively improves this aspect of the analysis (but still requires a minimum separation). Remarkably, by focusing attention on the local region around the true parameters, they obtain non-asymptotic bounds on the parameter estimation error. Our work is complementary to their results in that we focus on asymptotic limits rather than finite sample analysis. This allows us to provide a global analysis of EM without separation or initialization conditions, which cannot be deduced from the results of Balakrishnan et al. or Klusowski and Brinda by taking limits.

Finally, two related works have appeared following the initial posting of this article (Xu et al., 2016). First, Daskalakis et al. (2016) concurrently and independently proved a convergence result comparable to our Theorem 1 for Model 1; for this case, they also provide an explicit rate of linear convergence. Second, Jin et al. (2016) show that similar results do not hold in general for uniform mixtures of three or more spherical Gaussian distributions: common initialization schemes for (Population or Sample-based) EM may lead to local maxima that are arbitrarily far from the global maximizer. Similar results were well-known for Lloyd's algorithm, but were not previously established for Population EM (Srebro, 2007).

## 2 Analysis of EM for Mixtures of Two Gaussians

In this section, we present our results for Population EM and Sample-based EM under both Model 1 and Model 2, and also discuss further implications about the expected log-likelihood function. Without loss of generality, we may assume that the known covariance matrix $\boldsymbol{\Sigma}$ is the identity matrix $\boldsymbol{I}_d$. Throughout, we denote the Euclidean norm by $\|\cdot\|$, and the signum function by $\mathrm{sgn}(\cdot)$ (where $\mathrm{sgn}(0) = 0$, $\mathrm{sgn}(z) = 1$ if $z > 0$, and $\mathrm{sgn}(z) = -1$ if $z < 0$).

### 2.1 Main Results for Population EM

We present results for Population EM for both models, starting with Model 1.

**Theorem 1.** *Assume $\boldsymbol{\theta}^{\star} \in \mathbb{R}^d \setminus \{\mathbf{0}\}$. Let $(\boldsymbol{\theta}^{\langle t \rangle})_{t \geq 0}$ denote the Population EM iterates for Model 1, and suppose $\langle \boldsymbol{\theta}^{\langle 0 \rangle}, \boldsymbol{\theta}^{\star} \rangle \neq 0$. There exists $\kappa_\theta \in (0, 1)$—depending only on $\boldsymbol{\theta}^{\star}$ and $\boldsymbol{\theta}^{\langle 0 \rangle}$—such that*

$$\left\| \boldsymbol{\theta}^{\langle t+1 \rangle} - \mathrm{sgn}(\langle \boldsymbol{\theta}^{\langle 0 \rangle}, \boldsymbol{\theta}^{\star} \rangle) \boldsymbol{\theta}^{\star} \right\| \leq \kappa_\theta \cdot \left\| \boldsymbol{\theta}^{\langle t \rangle} - \mathrm{sgn}(\langle \boldsymbol{\theta}^{\langle 0 \rangle}, \boldsymbol{\theta}^{\star} \rangle) \boldsymbol{\theta}^{\star} \right\|.$$

The proof of Theorem 1 and all other omitted proofs are given in the full version of this article (Xu et al., 2016). Theorem 1 asserts that if $\boldsymbol{\theta}^{\langle 0 \rangle}$ is not on the hyperplane $\{\boldsymbol{x} \in \mathbb{R}^d : \langle \boldsymbol{x}, \boldsymbol{\theta}^{\star} \rangle = 0\}$, then the sequence $(\boldsymbol{\theta}^{\langle t \rangle})_{t \geq 0}$ converges to either $\boldsymbol{\theta}^{\star}$ or $-\boldsymbol{\theta}^{\star}$.

Our next result shows that if $\langle \boldsymbol{\theta}^{\langle 0 \rangle}, \boldsymbol{\theta}^{\star} \rangle = 0$, then $(\boldsymbol{\theta}^{\langle t \rangle})_{t \geq 0}$ still converges, albeit to $\mathbf{0}$.

**Theorem 2.** *Let $(\boldsymbol{\theta}^{\langle t \rangle})_{t \geq 0}$ denote the Population EM iterates for Model 1. If $\langle \boldsymbol{\theta}^{\langle 0 \rangle}, \boldsymbol{\theta}^\star \rangle = 0$, then*

$$\boldsymbol{\theta}^{\langle t \rangle} \quad \to \quad \mathbf{0} \quad as \ t \to \infty \,.$$

Theorems 1 and 2 together characterize the fixed points of Population EM for Model 1, and fully specify the conditions under which each fixed point is reached. The results are simply summarized in the following corollary.

**Corollary 1.** *If $(\boldsymbol{\theta}^{\langle t \rangle})_{t \geq 0}$ denote the Population EM iterates for Model 1, then*

$$\boldsymbol{\theta}^{\langle t \rangle} \quad \to \quad \mathrm{sgn}(\langle \boldsymbol{\theta}^{\langle 0 \rangle}, \boldsymbol{\theta}^\star \rangle)\boldsymbol{\theta}^\star \quad as \ t \to \infty \,.$$

We now discuss Population EM with Model 2. To state our results more concisely, we use the following re-parameterization of the model parameters and Population EM iterates:

$$\boldsymbol{a}^{\langle t \rangle} \triangleq \frac{\boldsymbol{\mu}_1^{\langle t \rangle} + \boldsymbol{\mu}_2^{\langle t \rangle}}{2} - \frac{\boldsymbol{\mu}_1^\star + \boldsymbol{\mu}_2^\star}{2}, \qquad \boldsymbol{b}^{\langle t \rangle} \triangleq \frac{\boldsymbol{\mu}_2^{\langle t \rangle} - \boldsymbol{\mu}_1^{\langle t \rangle}}{2}, \qquad \boldsymbol{\theta}^\star \triangleq \frac{\boldsymbol{\mu}_2^\star - \boldsymbol{\mu}_1^\star}{2}. \qquad (11)$$

If the sequence of Population EM iterates $((\boldsymbol{\mu}_1^{\langle t \rangle}, \boldsymbol{\mu}_2^{\langle t \rangle}))_{t \geq 0}$ converges to $(\boldsymbol{\mu}_1^\star, \boldsymbol{\mu}_2^\star)$, then we expect $\boldsymbol{b}^{\langle t \rangle} \to \boldsymbol{\theta}^\star$. Hence, we also define $\beta^{\langle t \rangle}$ as the angle between $\boldsymbol{b}^{\langle t \rangle}$ and $\boldsymbol{\theta}^\star$, i.e.,

$$\beta^{\langle t \rangle} \quad \triangleq \quad \arccos \left( \frac{\langle \boldsymbol{b}^{\langle t \rangle}, \boldsymbol{\theta}^\star \rangle}{\|\boldsymbol{b}^{\langle t \rangle}\|\|\boldsymbol{\theta}^\star\|} \right) \quad \in \quad [0, \pi] \,.$$

(This is well-defined as long as $\boldsymbol{b}^{\langle t \rangle} \neq \mathbf{0}$ and $\boldsymbol{\theta}^\star \neq \mathbf{0}$.)

We first present results on Population EM with Model 2 under the initial condition $\langle \boldsymbol{b}^{\langle 0 \rangle}, \boldsymbol{\theta}^\star \rangle \neq 0$.

**Theorem 3.** *Assume $\boldsymbol{\theta}^\star \in \mathbb{R}^d \setminus \{\mathbf{0}\}$. Let $(\boldsymbol{a}^{\langle t \rangle}, \boldsymbol{b}^{\langle t \rangle})_{t \geq 0}$ denote the (re-parameterized) Population EM iterates for Model 2, and suppose $\langle \boldsymbol{b}^{\langle 0 \rangle}, \boldsymbol{\theta}^\star \rangle \neq 0$. Then $\boldsymbol{b}^{\langle t \rangle} \neq \mathbf{0}$ for all $t \geq 0$. Furthermore, there exist $\kappa_a \in (0, 1)$—depending only on $\|\boldsymbol{\theta}^\star\|$ and $|\langle \boldsymbol{b}^{\langle 0 \rangle}, \boldsymbol{\theta}^\star \rangle / \|\boldsymbol{b}^{\langle 0 \rangle}\||$—and $\kappa_\beta \in (0, 1)$—depending only on $\|\boldsymbol{\theta}^\star\|$, $\langle \boldsymbol{b}^{\langle 0 \rangle}, \boldsymbol{\theta}^\star \rangle / \|\boldsymbol{b}^{\langle 0 \rangle}\|$, $\|\boldsymbol{a}^{\langle 0 \rangle}\|$, and $\|\boldsymbol{b}^{\langle 0 \rangle}\|$—such that*

$$\|\boldsymbol{a}^{\langle t+1 \rangle}\|^2 \quad \leq \quad \kappa_a^2 \cdot \|\boldsymbol{a}^{\langle t \rangle}\|^2 + \frac{\|\boldsymbol{\theta}^\star\|^2 \sin^2(\beta^{\langle t \rangle})}{4} \,,$$
$$\sin(\beta^{\langle t+1 \rangle}) \quad \leq \quad \kappa_\beta^t \cdot \sin(\beta^{\langle 0 \rangle}) \,.$$

By combining the two inequalities from Theorem 3, we conclude

$$\|\boldsymbol{a}^{\langle t+1 \rangle}\|^2 \quad = \quad \kappa_a^{2t}\|\boldsymbol{a}^{\langle 0 \rangle}\|^2 + \frac{\|\boldsymbol{\theta}^\star\|^2}{4} \sum_{\tau=0}^{t} \kappa_a^{2\tau} \cdot \sin^2(\beta^{\langle t-\tau \rangle})$$

$$\leq \quad \kappa_a^{2t}\|\boldsymbol{a}^{\langle 0 \rangle}\|^2 + \frac{\|\boldsymbol{\theta}^\star\|^2}{4} \sum_{\tau=0}^{t} \kappa_a^{2\tau} \kappa_\beta^{2(t-\tau)} \cdot \sin^2(\beta^{\langle 0 \rangle})$$

$$\leq \quad \kappa_a^{2t}\|\boldsymbol{a}^{\langle 0 \rangle}\|^2 + \frac{\|\boldsymbol{\theta}^\star\|^2}{4} t \left( \max\{\kappa_a, \kappa_\beta\} \right)^t \sin^2(\beta^{\langle 0 \rangle}) \,.$$

Theorem 3 shows that the re-parameterized Population EM iterates converge, at a linear rate, to the average of the two means $(\boldsymbol{\mu}_1^\star + \boldsymbol{\mu}_2^\star)/2$, as well as the line spanned by $\boldsymbol{\theta}^\star$. The theorem, however, does not provide any information on the convergence of the magnitude of $\boldsymbol{b}^{\langle t \rangle}$ to the magnitude of $\boldsymbol{\theta}^\star$. This is given in the next theorem.

**Theorem 4.** *Assume $\boldsymbol{\theta}^\star \in \mathbb{R}^d \setminus \{\mathbf{0}\}$. Let $(\boldsymbol{a}^{\langle t \rangle}, \boldsymbol{b}^{\langle t \rangle})_{t \geq 0}$ denote the (re-parameterized) Population EM iterates for Model 2, and suppose $\langle \boldsymbol{b}^{\langle 0 \rangle}, \boldsymbol{\theta}^\star \rangle \neq 0$. Then there exist $T_0 > 0$, $\kappa_b \in (0, 1)$, and $c_b > 0$—all depending only on $\|\boldsymbol{\theta}^\star\|$, $|\langle \boldsymbol{b}^{\langle 0 \rangle}, \boldsymbol{\theta}^\star \rangle / \|\boldsymbol{b}^{\langle 0 \rangle}\||$, $\|\boldsymbol{a}^{\langle 0 \rangle}\|$, and $\|\boldsymbol{b}^{\langle 0 \rangle}\|$—such that*

$$\left\| \boldsymbol{b}^{\langle t+1 \rangle} - \mathrm{sgn}(\langle \boldsymbol{b}^{\langle 0 \rangle}, \boldsymbol{\theta}^\star \rangle)\boldsymbol{\theta}^\star \right\|^2 \quad \leq \quad \kappa_b^2 \cdot \left\| \boldsymbol{b}^{\langle t \rangle} - \mathrm{sgn}(\langle \boldsymbol{b}^{\langle 0 \rangle}, \boldsymbol{\theta}^\star \rangle)\boldsymbol{\theta}^\star \right\|^2 + c_b \cdot \|\boldsymbol{a}^{\langle t \rangle}\| \quad \forall t > T_0 \,.$$

If $\langle \boldsymbol{b}^{\langle 0 \rangle}, \boldsymbol{\theta}^\star \rangle = 0$, then we show convergence of the (re-parameterized) Population EM iterates to the degenerate solution $(\boldsymbol{0}, \boldsymbol{0})$.

**Theorem 5.** *Let $(\boldsymbol{a}^{\langle t \rangle}, \boldsymbol{b}^{\langle t \rangle})_{t \geq 0}$ denote the (re-parameterized) Population EM iterates for Model 2. If $\langle \boldsymbol{b}^{\langle 0 \rangle}, \boldsymbol{\theta}^\star \rangle = 0$, then*

$$(\boldsymbol{a}^{\langle t \rangle}, \boldsymbol{b}^{\langle t \rangle}) \quad \rightarrow \quad (\boldsymbol{0}, \boldsymbol{0}) \quad as\ t \to \infty\,.$$

Theorems 3, 4, and 5 together characterize the fixed points of Population EM for Model 2, and fully specify the conditions under which each fixed point is reached. The results are simply summarized in the following corollary.

**Corollary 2.** *If $(\boldsymbol{a}^{\langle t \rangle}, \boldsymbol{b}^{\langle t \rangle})_{t \geq 0}$ denote the (re-parameterized) Population EM iterates for Model 2, then*

$$\boldsymbol{a}^{\langle t \rangle} \quad \rightarrow \quad \frac{\boldsymbol{\mu}_1^\star + \boldsymbol{\mu}_2^\star}{2} \quad as\ t \to \infty\,,$$

$$\boldsymbol{b}^{\langle t \rangle} \quad \rightarrow \quad \mathrm{sgn}(\langle \boldsymbol{b}^{\langle 0 \rangle}, \boldsymbol{\mu}_2^\star - \boldsymbol{\mu}_1^\star \rangle)\frac{\boldsymbol{\mu}_2^\star - \boldsymbol{\mu}_1^\star}{2} \quad as\ t \to \infty\,.$$

## 2.2 Main Results for Sample-based EM

Using the results on Population EM presented in the above section, we can now establish consistency of (Sample-based) EM. We focus attention on Model 2, as the same results for Model 1 easily follow as a corollary. First, we state a simple connection between the Population EM and Sample-based EM iterates.

**Theorem 6.** *Suppose Population EM and Sample-based EM for Model 2 have the same initial parameters: $\hat{\boldsymbol{\mu}}_1^{\langle 0 \rangle} = \boldsymbol{\mu}_1^{\langle 0 \rangle}$ and $\hat{\boldsymbol{\mu}}_2^{\langle 0 \rangle} = \boldsymbol{\mu}_2^{\langle 0 \rangle}$. Then for each iteration $t \geq 0$,*

$$\hat{\boldsymbol{\mu}}_1^{\langle t \rangle} \ \rightarrow \ \boldsymbol{\mu}_1^{\langle t \rangle} \quad and \quad \hat{\boldsymbol{\mu}}_2^{\langle t \rangle} \ \rightarrow \ \boldsymbol{\mu}_2^{\langle t \rangle} \quad as\ n \to \infty\,,$$

*where convergence is in probability.*

Note that Theorem 6 does not necessarily imply that the fixed point of Sample-based EM (when initialized at $(\hat{\boldsymbol{\mu}}_1^{\langle 0 \rangle}, \hat{\boldsymbol{\mu}}_2^{\langle 0 \rangle}) = (\boldsymbol{\mu}_1^{\langle 0 \rangle}, \boldsymbol{\mu}_2^{\langle 0 \rangle})$) is the same as that of Population EM. It is conceivable that as $t \to \infty$, the discrepancy between (the iterates of) Sample-based EM and Population EM increases. We show that this is not the case: the fixed points of Sample-based EM indeed converge to the fixed points of Population EM.

**Theorem 7.** *Suppose Population EM and Sample-based EM for Model 2 have the same initial parameters: $\hat{\boldsymbol{\mu}}_1^{\langle 0 \rangle} = \boldsymbol{\mu}_1^{\langle 0 \rangle}$ and $\hat{\boldsymbol{\mu}}_2^{\langle 0 \rangle} = \boldsymbol{\mu}_2^{\langle 0 \rangle}$. If $\langle \boldsymbol{\mu}_2^{\langle 0 \rangle} - \boldsymbol{\mu}_1^{\langle 0 \rangle}, \boldsymbol{\theta}^\star \rangle \neq 0$, then*

$$\limsup_{t \to \infty} |\hat{\boldsymbol{\mu}}_1^{\langle t \rangle} - \boldsymbol{\mu}_1^{\langle t \rangle}| \ \rightarrow \ 0 \quad and \quad \limsup_{t \to \infty} |\hat{\boldsymbol{\mu}}_2^{\langle t \rangle} - \boldsymbol{\mu}_2^{\langle t \rangle}| \ \rightarrow \ 0 \quad as\ n \to \infty\,,$$

*where convergence is in probability.*

## 2.3 Population EM and Expected Log-likelihood

Do the results we derived in the last section regarding the performance of EM provide any information on the performance of other ascent algorithms, such as gradient ascent, that aim to maximize the log-likelihood function? To address this question, we show how our analysis can determine the stationary points of the expected log-likelihood and characterize the shape of the expected log-likelihood in a neighborhood of the stationary points. Let $G(\boldsymbol{\eta})$ denote the expected log-likelihood, i.e.,

$$G(\boldsymbol{\eta}) \triangleq \mathbb{E}(\log f_{\boldsymbol{\eta}}(\boldsymbol{Y})) = \int f(\boldsymbol{y}; \boldsymbol{\eta}^*) \log f(\boldsymbol{y}; \boldsymbol{\eta})\, \mathrm{d}\boldsymbol{y},$$

where $\boldsymbol{\eta}^*$ denotes the true parameter value. Also consider the following standard regularity conditions:

**R1** The family of probability density functions $f(\boldsymbol{y}; \boldsymbol{\eta})$ have common support.

**R2** $\nabla_{\boldsymbol{\eta}} \int f(\boldsymbol{y}; \boldsymbol{\eta}^*) \log f(\boldsymbol{y}; \boldsymbol{\eta})\, \mathrm{d}\boldsymbol{y} = \int f(\boldsymbol{y}; \boldsymbol{\eta}^*) \nabla_{\boldsymbol{\eta}} \log f(\boldsymbol{y}; \boldsymbol{\eta})\, \mathrm{d}\boldsymbol{y}$, where $\nabla_{\boldsymbol{\eta}}$ denotes the gradient with respect to $\boldsymbol{\eta}$.

**R3** $\nabla_{\boldsymbol{\eta}}(\mathbb{E}\sum_{\boldsymbol{z}} f(\boldsymbol{z} \mid \boldsymbol{Y}; \boldsymbol{\eta}^{\langle t \rangle})) \log f(\boldsymbol{Y}, \boldsymbol{z}; \eta) = \mathbb{E}\sum_{\boldsymbol{z}} f(\boldsymbol{z} \mid \boldsymbol{Y}; \boldsymbol{\eta}^{\langle t \rangle}) \nabla_{\boldsymbol{\eta}} \log f(\boldsymbol{Y}, \boldsymbol{z}; \boldsymbol{\eta})$.

These conditions can be easily confirmed for many models including the Gaussian mixture models. The following theorem connects the fixed points of the Population EM and the stationary points of the expected log-likelihood.

**Lemma 1.** *Let $\bar{\boldsymbol{\eta}} \in \mathbb{R}^d$ denote a stationary point of $G(\boldsymbol{\eta})$. Also assume that $Q(\boldsymbol{\eta} \mid \boldsymbol{\eta}^{\langle t \rangle})$ has a unique and finite stationary point in terms of $\boldsymbol{\eta}$ for every $\boldsymbol{\eta}^{\langle t \rangle}$, and this stationary point is its global maxima. Then, if the model satisfies conditions R1–R3, and the Population EM algorithm is initialized at $\bar{\boldsymbol{\eta}}$, it will stay at $\bar{\boldsymbol{\eta}}$. Conversely, any fixed point of Population EM is a stationary point of $G(\boldsymbol{\eta})$.*

*Proof.* Let $\bar{\boldsymbol{\eta}}$ denote a stationary point of $G(\boldsymbol{\eta})$. We first prove that $\bar{\boldsymbol{\eta}}$ is a stationary point of $Q(\boldsymbol{\eta} \mid \bar{\boldsymbol{\eta}})$.

$$
\begin{aligned}
\nabla_{\boldsymbol{\eta}} Q(\boldsymbol{\eta} \mid \bar{\boldsymbol{\eta}})\big|_{\boldsymbol{\eta}=\bar{\boldsymbol{\eta}}} &= \int \sum_{\boldsymbol{z}} f(\boldsymbol{z} \mid \boldsymbol{y}; \bar{\boldsymbol{\eta}}) \frac{\nabla_{\boldsymbol{\eta}} f(\boldsymbol{y}, \boldsymbol{z}; \boldsymbol{\eta})\big|_{\boldsymbol{\eta}=\bar{\boldsymbol{\eta}}}}{f(\boldsymbol{y}, \boldsymbol{z}; \bar{\boldsymbol{\eta}})} f(\boldsymbol{y}; \boldsymbol{\eta}^*) \, \mathrm{d}\boldsymbol{y} \\
&= \int \sum_{\boldsymbol{z}} \frac{\nabla_{\boldsymbol{\eta}} f(\boldsymbol{y}, \boldsymbol{z}; \boldsymbol{\eta})\big|_{\boldsymbol{\eta}=\bar{\boldsymbol{\eta}}}}{f(\boldsymbol{y}; \bar{\boldsymbol{\eta}})} f(\boldsymbol{y}; \boldsymbol{\eta}^*) \, \mathrm{d}\boldsymbol{y} \\
&= \int \frac{\nabla_{\boldsymbol{\eta}} f(\boldsymbol{y}, \boldsymbol{\eta})\big|_{\boldsymbol{\eta}=\bar{\boldsymbol{\eta}}}}{f(\boldsymbol{y}; \bar{\boldsymbol{\eta}})} f(\boldsymbol{y}; \boldsymbol{\eta}^*) \, \mathrm{d}\boldsymbol{y} = \boldsymbol{0} \,,
\end{aligned}
$$

where the last equality is using the fact that $\bar{\boldsymbol{\eta}}$ is a stationary point of $G(\boldsymbol{\eta})$. Since $Q(\boldsymbol{\eta} \mid \bar{\boldsymbol{\eta}})$ has a unique stationary point, and we have assumed that the unique stationary point is its global maxima, then Population EM will stay at that point. The proof of the other direction is similar. $\square$

**Remark 1.** *The fact that $\boldsymbol{\eta}^*$ is the global maximizer of $G(\boldsymbol{\eta})$ is well-known in the statistics and machine learning literature (e.g., Conniffe, 1987). Furthermore, the fact that $\boldsymbol{\eta}^*$ is a global maximizer of $Q(\boldsymbol{\eta} \mid \boldsymbol{\eta}^*)$ is known as the self-consistency property (Balakrishnan et al., 2014).*

It is straightforward to confirm the conditions of Lemma 1 for mixtures of Gaussians. This lemma confirms that Population EM may be trapped in every local maxima. However, less intuitively it may get stuck at local minima or saddle points as well. Our next result characterizes the stationary points of $G(\boldsymbol{\theta})$ for Model 1.

**Corollary 3.** *$G(\boldsymbol{\theta})$ has only three stationary points. If $d = 1$ (so $\boldsymbol{\theta} = \theta \in \mathbb{R}$), then $0$ is a local minima of $G(\theta)$, while $\theta^*$ and $-\theta^*$ are global maxima. If $d > 1$, then $\boldsymbol{0}$ is a saddle point, and $\boldsymbol{\theta}^\star$ and $-\boldsymbol{\theta}^\star$ are global maxima.*

The proof is a straightforward result of Lemma 1 and Corollary 1. The phenomenon that Population EM may stuck in local minima or saddle points also happens in Model 2. We can employ Corollary 2 and Lemma 1 to explain the shape of the expected log-likelihood function $G$. To simplify the notation, we consider the re-parametrization $\boldsymbol{a} \triangleq \frac{\boldsymbol{\mu}_1 + \boldsymbol{\mu}_2}{2}$ and $\boldsymbol{b} \triangleq \frac{\boldsymbol{\mu}_2 - \boldsymbol{\mu}_1}{2}$.

**Corollary 4.** *$G(\boldsymbol{a}, \boldsymbol{b})$ has three stationary points:*

$$
\left( \frac{\boldsymbol{\mu}_1^\star + \boldsymbol{\mu}_2^\star}{2}, \frac{\boldsymbol{\mu}_2^\star - \boldsymbol{\mu}_1^\star}{2} \right), \qquad \left( \frac{\boldsymbol{\mu}_1^\star + \boldsymbol{\mu}_2^\star}{2}, \frac{\boldsymbol{\mu}_1^\star - \boldsymbol{\mu}_2^\star}{2} \right), \qquad and \qquad \left( \frac{\boldsymbol{\mu}_1^\star + \boldsymbol{\mu}_2^\star}{2}, \frac{\boldsymbol{\mu}_1^\star + \boldsymbol{\mu}_2^\star}{2} \right).
$$

*The first two points are global maxima. The third point is a saddle point.*

## 3 Concluding Remarks

Our analysis of Population EM and Sample-based EM shows that the EM algorithm can, at least for the Gaussian mixture models studied in this work, compute statistically consistent parameter estimates. Previous analyses of EM only established such results for specific methods of initializing EM (e.g., Dasgupta and Schulman, 2007; Balakrishnan et al., 2014); our results show that they are not really necessary in the large sample limit. However, in any real scenario, the large sample limit may not accurately characterize the behavior of EM. Therefore, these specific methods for initialization, as well as non-asymptotic analysis, are clearly still needed to understand and effectively apply EM.

There are several interesting directions concerning EM that we hope to pursue in follow-up work. The first considers the behavior of EM when the dimension $d = d_n$ may grow with the sample size

$n$. Our proof of Theorem 7 reveals that the parameter error of the $t$-th iterate (in Euclidean norm) is of the order $\sqrt{d/n}$ as $t \to \infty$. Therefore, we conjecture that the theorem still holds as long as $d_n = o(n)$. This would be consistent with results from statistical physics on the MLE for Gaussian mixtures, which characterize the behavior when $d_n \propto n$ as $n \to \infty$ (Barkai and Sompolinsky, 1994).

Another natural direction is to extend these results to more general Gaussian mixture models (e.g., with unequal mixing weights or unequal covariances) and other latent variable models.

**Acknowledgements.** The second named author thanks Yash Deshpande and Sham Kakade for many helpful initial discussions. JX and AM were partially supported by NSF grant CCF-1420328. DH was partially supported by NSF grant DMREF-1534910 and a Sloan Fellowship.

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
