[Reviews · NeurIPS 2016]

Reviewer 1

Summary

In two particular cases of mixtures of Gaussians distributions, the authors characterize the limits of a population EM algorithm in terms of the starting points. Then, they prove that the sample-based EM algorithm has the same asymptotic properties as the population EM with same starting points.

Qualitative Assessment

The focus of this work is the performance of the EM algorithm for two very specific models (mixture of two Gaussian distributions). The results are interesting and all cases are covered, hence, the authors provide a complete description of the performance of the EM in that setup. My only negative remark is that I believe that the proofs are unreasonably long, given that the paper focuses on a very specific model. However, mixtures of even two Gaussian components are not fully understood yet, so I am favorable for acceptance of this work at NIPS.

Confidence in this Review

2-Confident (read it all; understood it all reasonably well)


Reviewer 2

Summary

This paper proves that EM algorithm converges to the global minimum when applied to mean-estimation of mixture-of-Gaussian distributions.

Qualitative Assessment

The result is important and a great contribution to our understanding of EM. I regret starting this review too late to have enough time to read through the proofs; I will read it carefully over the next couple of days. Also, the result is, to the best of my knowledge, quite novel. I can't yet comment on the proof, but it looks involved and nontrivial. The result is beautiful as are the writing and the presentation. This paper is a pleasure to read.

Confidence in this Review

1-Less confident (might not have understood significant parts)


Reviewer 3

Summary

The authors prove global convergence of the EM algorithm in a Gaussian mixture model where the covariance and mixing proportions are known a priori.

Qualitative Assessment

The presentation quality is mostly good but I didn't get a good enough sense of what the authors intended the main selling point of the paper to be. I'm not an expert in this area but the article didn't give a clear enough sense of what the practical importance of its results are. Models 1-2 are easy enough to estimate with large samples using, e.g., method of moments (MOM) or MOM plus a one-step estimator. Presumably MOM, which is sqrt(n)-consistent, is also a good enough pilot estimator for the techniques of Balakrishnan et al. (2014) to apply. The paper might still be acceptable if the proof techniques were interesting or novel enough to possibly generalize to other, more difficult-to-estimate models. But in that case the proof techniques should have been explained in the main body of the paper, at least heuristically. As it is, they are in a very long technical supplement without much in the way of intuitive explanation. Also, it's hard for me to understand why these models were an interesting test case for the EM algorithm. For example, in Model 1 the population likelihood function has only two local maxima (the global maxima at theta and -theta) and one saddle point. Given the general properties of the EM algorithm as a majorization-maximization procedure, convergence to one of the global maxima seems like a foregone conclusion.

Confidence in this Review

2-Confident (read it all; understood it all reasonably well)


Reviewer 4

Summary

The authors provide a global analysis of expectation maximization for mixtures of two Gaussians, where both Gaussians have the same covariance matrix and the same weight in the mixture. First, global results in the population version is established, showing how EM would behave if the algorithm had access to the true underlying distributions. Next, based on the population analysis, the authors show the behavior of sample based EM is similar to that of population EM, using consistency type arguments.

Qualitative Assessment

I have a question regarding to why did the authors only provide a consistency-type result for the finite sample case? What are the reasons preventing the authors' analysis from being able to give finite sample global results for EM?

Confidence in this Review

2-Confident (read it all; understood it all reasonably well)


Reviewer 5

Summary

This article analyses Expectation Maximization (EM) algorithm on balanced mixtures of two Gaussians with known variance. The means are estimated. The questions of the rate of convergence and the accuracy of EM are addressed. The main tool is the analysis of Population EM. EM and Population EM stationary points are characterized.

Qualitative Assessment

The results are well presented and sufficiently commented. This paper has a practical impact since it considers the difference between the theory of the maximum likelihood estimate and its computation using EM. The focus is on asymptotic limits and is complementary to the literature. Remarks : - An other natural perspective would be to consider the estimation of both the means and the variances, and then a mixture of more than two Gaussians. - A world could have been said on the robustness of EM to model misspecification : what happens if the data are drawn from one Gaussian for example?

Confidence in this Review

3-Expert (read the paper in detail, know the area, quite certain of my opinion)


Reviewer 6

Summary

The authors provide an analysis of the EM algorithm for two toy Gaussian mixture models characterizing the initial parameter values that lead to convergence. To this end, they use the true Q function-- obtained as an expectation instead of doing a sample average-- to give the ideal parameter sequence from a given initial value. This sequence is used for the characterization of the initial parameter values that lead to the true parameter. They also demonstrate that the actual sequence of real run converges to the idealized sequence in probability. Later they use this analysis to explore the shape of the expected log likelihood in the neighborhood of its stationary points.

Qualitative Assessment

Congratulations to the authors for a good paper! Quality: The paper is technically sound with non trivial results. The conclusions of the paper are well supported by the theory. The condition on the initial parameter that leads to convergence to the the true parameter are particularly interesting. Clarity: This is a very well written paper and it reads well. The math, though not trivial, is very accessible because of the presentation. The authors have provided adequate commentary that aids intuition and understanding. Originality: The idea of the paper to analyze an idealized execution of the EM algorithm, by using the true Q function, instead of the one obtained by sample average is quite novel. The authors have provided adequate discussion of the relevant literature to inform the reader about the state of the art and evaluate their work in the right context. Significance: The results of the paper are very interesting, they help understand the sensitivity of EM (Gaussian mixtures) to initial parameters. Though the models considered are too restrictive for real world data, but they work more like a proof of concept for the analysis technique. It remains to see if the technique would work for more general models. Strengths: Solid analysis Weakness: Restrictive model. Comments: In line 188-192 authors claim that their analysis for EM provides understanding for other ascent algorithm. However, in the justification that follows the conclusions about G(\eta) are still made using the Q function, which is in the EM framework. It is not clear what the authors want to claim about other algorithms. Overall, I would recommend accepting the paper without reservations.

Confidence in this Review

2-Confident (read it all; understood it all reasonably well)